# Kinetochore protein depletion underlies cytokinesis failure and somatic polyploidization in the moss *Physcomitrella patens*

Elena Kozgunova[1]*, Momoko Nishina[2], Gohta Goshima[2]*

[1]International Collaborative Programme in Science, Graduate School of Science, Nagoya University, Nagoya, Japan; [2]Division of Biological Science, Graduate School of Science, Nagoya University, Nagoya, Japan

**Abstract** Lagging chromosome is a hallmark of aneuploidy arising from errors in the kinetochore–spindle attachment in animal cells. However, kinetochore components and cellular phenotypes associated with kinetochore dysfunction are much less explored in plants. Here, we carried out a comprehensive characterization of conserved kinetochore components in the moss *Physcomitrella patens* and uncovered a distinct scenario in plant cells regarding both the localization and cellular impact of the kinetochore proteins. Most surprisingly, knock-down of several kinetochore proteins led to polyploidy, not aneuploidy, through cytokinesis failure in >90% of the cells that exhibited lagging chromosomes for several minutes or longer. The resultant cells, containing two or more nuclei, proceeded to the next cell cycle and eventually developed into polyploid plants. As lagging chromosomes have been observed in various plant species in the wild, our observation raised a possibility that they could be one of the natural pathways to polyploidy in plants.

DOI: https://doi.org/10.7554/eLife.43652.001

*For correspondence:
kozgunova@gmail.com (EK);
goshima@bio.nagoya-u.ac.jp (GG)

**Competing interests:** The authors declare that no competing interests exist.

## Introduction

The kinetochore is a macromolecular complex that connects chromosomes to spindle microtubules and plays a central role in chromosome segregation. Kinetochore malfunction causes checkpoint-dependent mitotic arrest, apoptosis, and/or aneuploidy-inducing chromosome missegregation (*Potapova and Gorbsky, 2017*). Most of our knowledge on kinetochore function and impact on genome stability is derived from animal and yeast studies (*Musacchio and Desai, 2017*). Another major group of eukaryotes, plants, also possesses conserved kinetochore proteins (*Yu et al., 2000*; *van Hooff et al., 2017*; *Yamada and Goshima, 2017*). Although the localization and loss-of-function phenotype of some plant kinetochore proteins have been reported before (*Shin et al., 2018*; *Zhang et al., 2018*; *Wang et al., 2012*; *Caillaud et al., 2009*; *Komaki and Schnittger, 2017*; *Lermontova et al., 2013*; *Sandmann et al., 2017*; *Sato et al., 2005*; *Du and Dawe, 2007*; *Ogura et al., 2004*), the data are mostly obtained from fixed cells of specific tissues. No comprehensive picture of plant kinetochore protein dynamics and functions can be drawn as of yet. For example, 12 out of 16 components that form CCAN (*c*onstitutive *c*entromere *a*ssociated *n*etwork) in animal and yeast cells cannot be identified by homology searches (*Musacchio and Desai, 2017*; *Yamada and Goshima, 2017*). How the residual four putative CCAN subunits act in plants is also unknown.

The moss *Physcomitrella patens* is an emerging model system for plant cell biology. The majority of its tissues are in a haploid state, and, owing to an extremely high rate of homologous

**eLife digest** Plants and animals, like all living things, are made of self-contained units called cells that are able to grow and multiply as required. Each cell contains structures called chromosomes that provide the genetic instructions needed to perform every task in the cell. When a cell is preparing to divide to make two identical daughter cells – a process called mitosis – it first needs to duplicate its chromosomes and separate them into two equal-sized sets. This process is carried out by complex cell machinery known as the spindle.

Structures called kinetochores assemble on the chromosomes to attach them to the spindle. Previous studies in animal cells have shown that, if the kinetochores do not work properly, one or more chromosomes may be left behind when the spindle operates. These 'lagging' chromosomes may ultimately land up in the wrong daughter cell, resulting in one of the cells having more chromosomes than the other. This can lead to cancer or other serious diseases in animals. However, it was not known what happens in plant cells when kinetochores fail to work properly.

To address this question, Kozgunova et al. used a technique called RNA interference (or RNAi for short) to temporarily interrupt the production of kinetochores in the cells of a moss called *Physcomitrella patens*. Unexpectedly, the experiments found that most of the moss cells with lagging chromosomes were unable to divide. Instead, they remained as single cells that had twice the number of chromosomes as normal, a condition known as polyploidy. After the effects of the RNAi wore off, these polyploid moss cells were able to divide normally and were successfully grown into moss plants with a polyploid number of chromosomes.

Polyploidy is actually widespread in the plant kingdom, and it has major impacts on plant evolution. It is also known to increase the amount of food that crops produce. However, it is still unclear why polyploidy is so common in plants. By showing that errors in mitosis may also be able to double the number of chromosomes in plant cells, the findings of Kozgunova et al. provide new insights into plant evolution and, potentially, a method to increase polyploidy in crop plants in the future.

DOI: https://doi.org/10.7554/eLife.43652.002

recombination, gene disruption and fluorescent protein tagging of endogenous genes are easy to obtain in the first generation (*Cove et al., 2006*). The homology search indicated that all the *P. patens* proteins identified as the homologue of human kinetochore components are conserved in the most popular model plant species *A. thaliana* (*Yamada and Goshima, 2017*): therefore, the knowledge gained in *P. patens* would be largely applicable to flowering plants, including crop species. Another remarkable feature of *P. patens* is its regeneration ability; for example, differentiated gametophore leaf cells, when excised, are efficiently reprogrammed to become stem cells (*Sato et al., 2017*; *Ishikawa et al., 2011*). Thus, genome alteration even in a somatic cell can potentially spread through the population.

In this study, we aimed to comprehensively characterize conserved kinetochore proteins in a single-cell type, the *P. patens* caulonemal apical cell. We observed that many proteins displayed localization patterns distinct from their animal counterparts. Furthermore, kinetochore malfunction led to chromosome missegregation and microtubule disorganization in the phragmoplast, eventually resulting in cytokinesis failure and polyploidy.

## Results

### Endogenous localization analysis of conserved kinetochore proteins in *P. patens*

To observe the endogenous localization of putative kinetochore components, we inserted a fluorescent tag in-frame at the N- and/or C-terminus of 18 selected proteins, which contain at least one subunit per sub-complex (*Figure 1—figure supplement 1*). Initially, we conducted C-terminal tagging since the success rate of homologous recombination is much higher than N-terminal tagging (*Yamada et al., 2016*). For 10 proteins, function was unlikely perturbed by tagging, as the transgenic moss grew indistinguishably from wild-type, despite the single-copy protein being replaced with the

tagged protein. For other proteins, the functionality of the tagged version could not be verified, since untagged paralogs are present in the genome. The C-terminal tagging line for CENP-S could not be obtained after two attempts, suggesting that tagging affected the protein's function and thereby moss viability. The N-termini of CENP-S, CENP-O, and CENP-C were also tagged with Citrine. Among them, no paralogous protein could be identified for CENP-C; therefore, Citrine signals would precisely represent the endogenous localization. Exceptionally, histone H3-like CENP-A (CenH3) localization was determined by ectopic Citrine-CENP-A expression, as tagging likely perturbs its function.

Consistent with their sequence homology, many of the proteins were localized to the kinetochore at least transiently during the cell cycle. However, multiple proteins also showed unexpected localization (or disappearance) at certain cell cycle stages (*Figure 1—figure supplement 2–7*; *Videos 1–4*). Most surprising were CCAN protein dynamics: CENP-X, CENP-O and CENP-S did not show kinetochore enrichment at any stages (*Figure 1—figure supplement 3*; *Videos 1* and *3*), whereas CENP-C also dissociated from the kinetochore transiently in the post-mitotic phase (*Figure 1B—Figure 1—figure supplement 4*; *Videos 2* and *3*). Thus, we could not identify any 'constitutive' kinetochore proteins other than CENP-A.

## Kinetochore malfunction causes chromosome missegregation and cytokinesis failure

We failed to obtain knockout lines and/or induce frameshift mutation using CRISPR/Cas9 for the single-copy kinetochore proteins, except for the spindle checkpoint protein Mad2, strongly suggesting that they are essential for moss viability. We therefore made conditional RNAi lines, targeting different proteins from both inner and outer kinetochores (summarized in *Figure 1—figure supplement 1*). In this RNAi system, knockdown of target genes was induced by the addition of β-estradiol to the culture medium 4–6 days prior to live-imaging (*Nakaoka et al., 2012*). Since RNAi sometimes exhibits an off-target effect, we prepared two independent RNAi constructs for most target genes. Following the previously established protocol (*Nakaoka et al., 2012*; *Miki et al., 2016*), we screened for cell growth/division phenotypes in ≥10 transgenic lines for each construct by using long-term (>10 hr) fluorescent imaging. We observed mitotic defects in multiple RNAi lines, such as delay in mitotic progression, chromosome missegregation and/or multi-nuclei; these phenotypes were never observed in the control line (*Figure 2A,B*; *Video 5*). A full list of targeted genes and brief descriptions of the observed phenotypes are provided in *Figure 1—figure supplement 1*.

We first selected CENP-A for detailed analysis, the only constitutive centromeric protein identified in *P. patens*. As expected, we observed a significant mitotic delay and chromosome alignment/segregation defects in the CENP-A RNAi lines (*Figure 2—figure supplement 1*; *Video 6*). These phenotypes can be explained by a deficiency in proper kinetochore-microtubule attachment. Consequently,

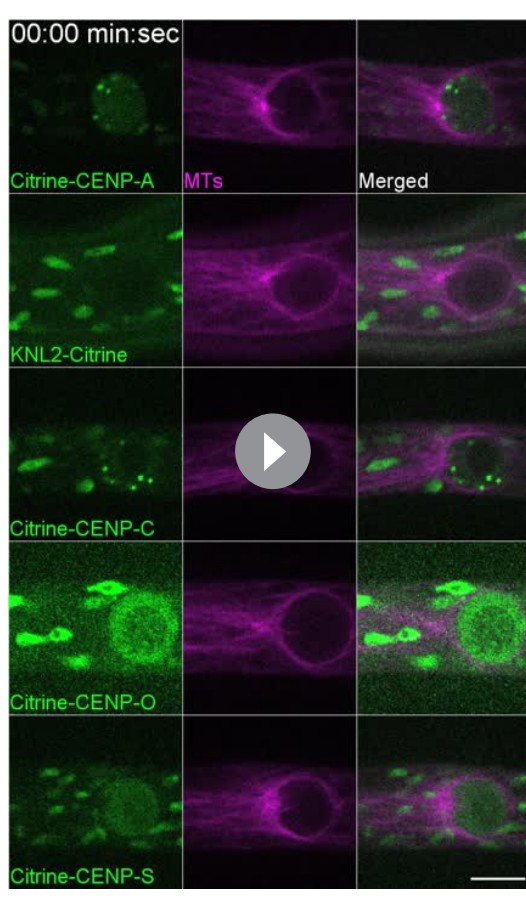

**Video 1.** Localization of the centromere and CCAN proteins during cell division. Live-cell imaging was conducted in *P. patens* protonemal cells expressing mCherry-tubulin (magenta) and one of the following tagged proteins(green): Citrine-CENP-A, KNL2-Citrine, Citrine-CENP-C, Citrine-CENP-O and Citrine-CENP-S. Note that brightness/contrast of Citrine-CENP-O images have been enhanced. Images are single focal plane and were acquired every 30 s. Bar, 10 μm.
DOI: https://doi.org/10.7554/eLife.43652.011

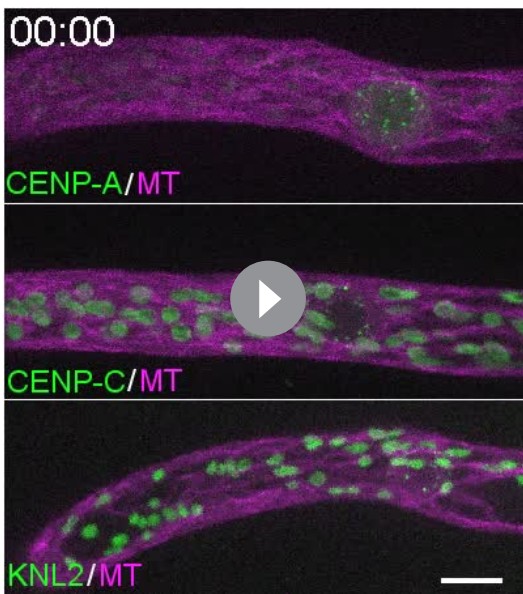

**Video 2.** Transient disappearance of CENP-C from the kinetochore after cell division. Live-cell imaging was conducted in *P. patens* protonemal cells expressing mCherry-tubulin (magenta) and one of the following tagged proteins (green): Citrine-CENP-A, Citrine-CENP-C and KNL2-Citrine. Displayed are the the merged images of a single focal plane for mCherry-tubulin (magenta) and maximum-projection of the Z-stack for Citrine-tagged proteins. Images were acquired every 5 min. Bar, 10 μm.
DOI: https://doi.org/10.7554/eLife.43652.012

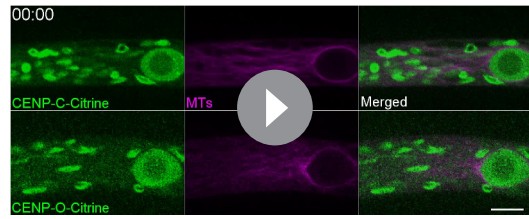

**Video 3.** Localization of the C-termini tagged CENP-C and CENP-O. Live-cell imaging was conducted in *P. patens* protonemal cells expressing mCherry-tubulin (magenta) and one of the following tagged proteins (green):CENP-C-Citrine and CENP-O-Citrine. Images are single focal plane and were acquired every 30 s. Bar, 10 μm.
DOI: https://doi.org/10.7554/eLife.43652.013

micronuclei were occasionally observed in the daughter cells, a hallmark of aneuploidy. We concluded that CENP-A, like in many organisms, is essential for equal chromosome segregation during mitosis in moss.

Surprisingly, we also frequently observed cells with two large nuclei in both RNAi lines (*Figures 2B* at 1 h 18 min), which is the typical outcome of cytokinesis failure in this cell type (*Kosetsu et al., 2013*; *Miki et al., 2014*; *Naito and Goshima, 2015*). To check if a similar phenotype is observed after the depletion of another kinetochore protein, we observed conditional RNAi line for SKA1, an outermost kinetochore component that does not directly interact with CENP-A and that had not been functionally characterized in the plant cells yet. As expected, mitotic delay and chromosome missegregation were observed in the RNAi line (*Figure 2B*–*Figure 2—figure supplement 1*; *Video 5*). In addition, cytokinesis failure was also detected (*Figure 2B*; *Video 7*). To verify that the observed phenotype of SKA1 was not due to an off-target effect, we ectopically expressed RNAi-insensitive SKA1-Cerulean in the RNAi line and observed the rescue of all the above phenotypes (*Figure 2—figure supplement 2*). Furthermore, we observed a similar phenotype in RNAi lines targeting CENP-C (CCAN), Nnf1 (Mis12 complex), KNL1 and Nuf2 (Ndc80 complex), suggesting that cytokinesis failure is a common outcome following kinetochore malfunction (*Figure 2*; *Video 5*).

Although we could not detect any kinetochore enrichment of the CCAN subunit CENP-X, we analyzed its RNAi lines. Interestingly, we observed similar phenotypes to CENP-A and SKA1, including cytokinesis failure (*Figure 2B*–*Figure 2—figure supplement 1*; *Video 6*). CENP-X RNAi phenotypes were rescued by the ectopic expression of CENP-X-Cerulean that was resistant to the RNAi construct (*Figure 2—figure supplement 2*). Thus, CENP-X has lost its kinetochore localization in moss, but is still essential for chromosome segregation and cell division.

By analyzing a total of 44 cells from SKA1 (9 cells), CENP-X (18 cells) and CENP-A RNAi (9 cells for one construct and 8 cells for the other) lines that had lagging chromosomes, we noticed a correlation between cytokinesis failure and lagging chromosomes lingering for a relatively long time in the space between separated chromatids. We therefore quantified the duration of lagging chromosomes' residence in the midzone between separating chromatids following anaphase onset. Interestingly, a minor delay of chromosomes in the midzone (<4 min) never perturbed cytokinesis (100%, n = 9 for CENP-A, n = 4 for CENP-X and n = 3 for SKA1). By contrast, if we observed a longer delay of chromosome clearance from the midzone, even when only a single chromosome was detectable,

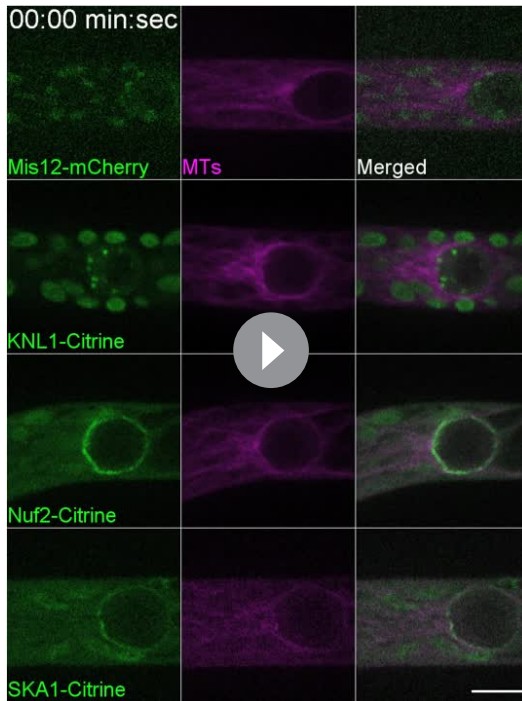

**Video 4.** Localization of the Mis12, KNL1, Nuf2 and SKA1 during cell division. Live-cell imaging was conducted in *P. patens* protonemal cells expressing mCherry-tubulin or GFP-tubulin (magenta) and one of the following tagged proteins: Mis12-mCherry, KNL1-Citrine, Nuf2-Citrine and SKA1-Citrine Images were acquired at a single focal plane every 30 s. Bar, 10 µm.
DOI: https://doi.org/10.7554/eLife.43652.014

cytokinesis defects occurred in 96% of the cells (n = 9, 14 and 5; *Figure 2C,D*—*Figure 2—source data 1*).

During plant cytokinesis, a bipolar microtubule-based structure known as the phragmoplast is assembled between segregating chromatids. The cell plate then forms in the phragmoplast midzone (~4 min after anaphase onset in *P. patens* caulonemal cells) and gradually expands toward the cell cortex, guided by the phragmoplast (*Kosetsu et al., 2013*). We observed that microtubules reorganized into phragmoplast-like structures upon chromosome segregation in every cell, regardless of the severity of chromosome missegregation (e.g. 32 min in *Figure 2B*). However, high-resolution imaging showed that microtubule interdigitates at the phragmoplast midzone were abnormal in the kinetochore RNAi lines. In 5 out of 7 control cells, a sharp microtubule overlap indicated by bright GFP-tubulin signals was observed during cytokinesis, as expected from previous studies (*Kosetsu et al., 2013*; *Hiwatashi et al., 2008*) (yellow arrowhead in *Figure 2E*). In contrast, CENP-A and SKA1 RNAi lines that had lagging chromosomes and eventually failed cytokinesis never exhibited such focused overlaps (0 out of 12 cells); instead, the overlap was broader and less distinguished (*Figure 2E*).

Finally, we checked if the cell plate was formed at any point in the cells that had cytokinesis defects, using the lipophilic FM4-64 dye. We could not observe vesicle fusion at the midzone following anaphase onset; thus, the cell plate did not form in the cells that had lagging chromosomes for a long time (*Figure 2B,E*; *Video 8*). From these results, we concluded that occupation of the midzone by lagging chromosomes for several minutes prevents proper phragmoplast assembly and cell plate formation, which subsequently causes cytokinesis failure.

## Polyploid plants are regenerated from isolated multi-nucleated cells

Lagging chromosomes are a major cause of aneuploidy in daughter cells, which is particularly deleterious for haploid cells. However, the above observation supports a different scenario, whereby cytokinesis failure induced by lagging chromosomes allows a cell to have a duplicated genome set in two or more nuclei. On the other hand, whether animal somatic cells that have failed cytokinesis can re-enter the cell cycle or not remains an ongoing debate (*Ganem and Pellman, 2007*; *Uetake and Sluder, 2004*; *Panopoulos et al., 2014*). To address whether moss cells can recover from severe cell division defects and continue their cell cycle, we first analyzed the DNA content of cells in the CENP-A exon-targeting RNAi line, in which multi-nucleated cells were most prevalent. For comparison, we used the parental line: the nuclei of anaphase/telophase cells served as the 1N reference and randomly selected interphase nuclei as the 2N reference, as caulonemal cells are mostly in the G2 phase (*Ishikawa et al., 2011*; *Schween et al., 2003*). We observed that the majority of the multinucleated cells after CENP-A RNAi underwent DNA replication and became tetraploid or attained even higher ploidy (*Figure 3A*; DNA was quantified at day 5 after β-estradiol treatment).

Next, we checked if multi-nucleated cells continue cell cycling. We used SKA1 RNAi line for a long (46 hr) time-lapse imaging; with this imaging, we expected to monitor the process of cytokinesis failure of a haploid cell and its fate. During the imaging period, we indeed observed cytokinesis failure and 10% or 25% multi-nucleated apical cells executed the next cell division by forming a

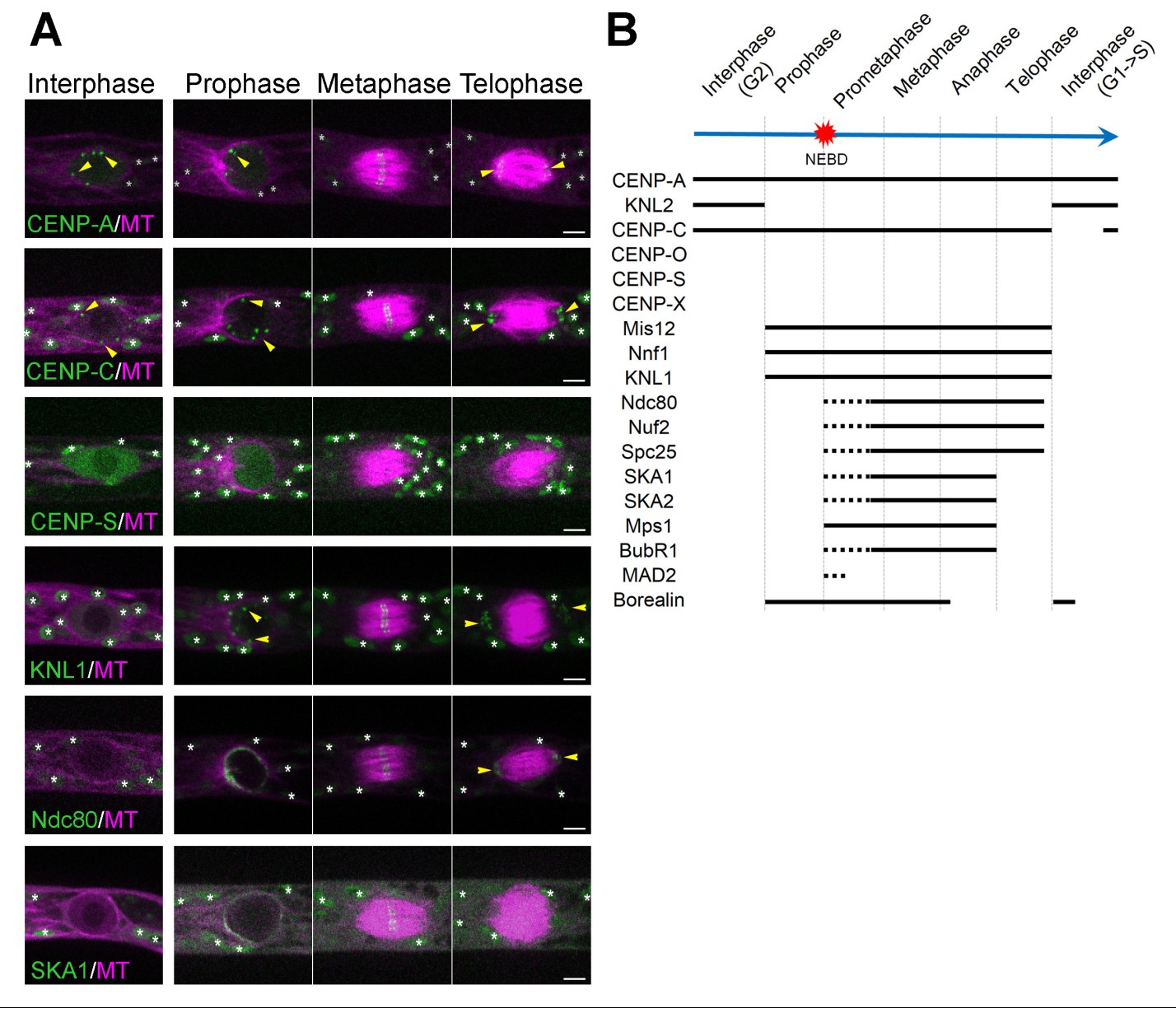

**Figure 1.** Unconventional localization of kinetochore proteins in *P. patens*. (**A**) Live imaging of *P. patens* caulonemal apical cells expressing mCherry-tubulin and selected kinetochore proteins: Citrine-CENP-A; Citrine-CENP-C; Citrine-CENP-S; KNL1-Citrine; Ndc80-Citrine and SKA1-Citrine. Full localization data can be found in Supplemental data. Some kinetochore signals are marked with yellow arrowheads, whereas autofluorescent chloroplasts are all marked with white asterisks. Images were acquired at a single focal plane. Bars, 5 μm. See *Figure 1—figure supplements 1–7*, *Videos 1–4*. (**B**) Timeline of centromere/kinetochore localization during the cell cycle in *P. patens* caulonemal apical cells. Solid lines correspond to the detection of clear kinetochore signals, whereas dotted lines indicate more dispersed signals.

DOI: https://doi.org/10.7554/eLife.43652.003

The following figure supplements are available for figure 1:

**Figure supplement 1.** Summary of kinetochore protein tagging and disruption/knockdown in *P. patens*.
DOI: https://doi.org/10.7554/eLife.43652.004

**Figure supplement 2.** Localization of CENP-A and KNL2/MIS18BP1 during cell division.
DOI: https://doi.org/10.7554/eLife.43652.005

**Figure supplement 3.** Localization of CCAN proteins during cell division.
DOI: https://doi.org/10.7554/eLife.43652.006

**Figure supplement 4.** CENP-C is not a constitutive centromeric protein in *P. patens*.
DOI: https://doi.org/10.7554/eLife.43652.007

*Figure 1 continued on next page*

*Figure 1 continued*

**Figure supplement 5.** Localization of Mis12, Nnf1 and KNL1 during cell division.
DOI: https://doi.org/10.7554/eLife.43652.008
**Figure supplement 6.** Localization of outer kinetochore proteins during cell division.
DOI: https://doi.org/10.7554/eLife.43652.009
**Figure supplement 7.** Localization of CPC and SAC proteins during cell division.
DOI: https://doi.org/10.7554/eLife.43652.010

single spindle (n = 43 and 25 for experiments 1 and 2, respectively, *Figure 3B*; *Video 9*). The reason for the low frequency of this event is unclear; strong chromosome missegregation might result in a severe 'aneuploid' state for each nucleus, whereas the cell is overall polyploid, which might change the cell physiology. Nevertheless, this data strongly suggests that cells that have undergone cytokinesis failure can continue cell cycling as diploids at a certain probability.

Diploid *P. patens* is known to develop protonema tissue with a few gametophores (leafy shoots) (*Schween et al., 2005*); therefore, a multi-nucleated cell produced by the cytokinesis failure of a caulonemal cell might proliferate and form a large protonema colony. To test this possibility, we isolated and cultured several cells (*Figure 3C*) that were seemingly multi-nuclear after SKA1 RNAi via laser dissection microscopy (note that there is an unambiguity in identifying multi-nucleate cells; see Materials and methods for detailed explanation). After 6 weeks of culturing without β-estradiol (i.e. RNAi was turned off), we obtained four moss colonies, two of which consisted mainly of protonemal cells with a few gametophores (*Figure 3D*, colony 3 and 4). DNA staining and quantification showed that the majority of the cells derived from those two colonies had DNA content approximately double of the control haploid cells, which were regenerated in an identical manner (*Figure 3E*, colony 3 and 4, regenerated from cell 3 and 4, respectively). Thus, a polyploid plant was regenerated from a single multi-nucleated somatic cell.

## Discussion

### Kinetochore protein dynamics in a plant cell

This study provides a comprehensive view of the dynamics of conserved kinetochore proteins in a single cell type of *P. patens*; furthermore, to the best of our knowledge, several proteins, including borealin, KNL1 and SKA subunits, have been characterized for the first time in plant cells. The tagged proteins were expressed under their native promoter at the original chromosome locus; thus, fluorescent signals of most, if not all, proteins would represent the endogenous localization.

Overall, the behavior of outer subunits was largely consistent with their animal counterparts, suggesting that the mitotic function is also conserved. However, the timing of kinetochore enrichment differed from that of animal cells and even flowering plants (e.g. Arabidopsis, maize) (*Shin et al., 2018*; *Du and Dawe, 2007*; *Hori et al., 2003*): for example, *P. patens* Ndc80 complex gradually accumulated at the kinetochore after NEBD, unlike Arabidopsis and maize, where it showed kinetochore enrichment throughout the cell cycle (*Shin et al., 2018*; *Du and Dawe, 2007*). More unexpected localizations were observed for inner CCAN subunits, namely CENP-C, CENP-O, CENP-S and CENP-X. For example, CENP-C disappeared from the centromeres shortly after mitotic exit. In animal cells, CENP-C has been suggested to act in cooperation with Mis18BP1/KNL2 to facilitate CENP-A deposition in late telophase and early G1 (2). Hence, the mechanism of CENP-A incorporation might have been modified in plants.

CENP-O, -S, or –X did not show kinetochore enrichment at any stage. CENP-X localization was unlikely an artifact of Citrine tagging, since the tagged protein rescued the RNAi phenotype. In human cells, 16 CCAN subunits, forming four sub-complexes, have been identified and shown to be critical for kinetochore assembly and function, not only in cells, but also in reconstitution systems (*Guse et al., 2011*; *Weir et al., 2016*). In plants, only four CCAN homologues have been identified through sequence homology search. It is therefore possible that less conserved CCAN subunits are present but could not be identified by the homology search. However, the complete lack of kinetochore localization for CENP-O, -S, -X suggests that plants have lost the entire kinetochore-enriched CCAN complex. Somewhat puzzlingly, CENP-X, despite its unusual localization, remained an

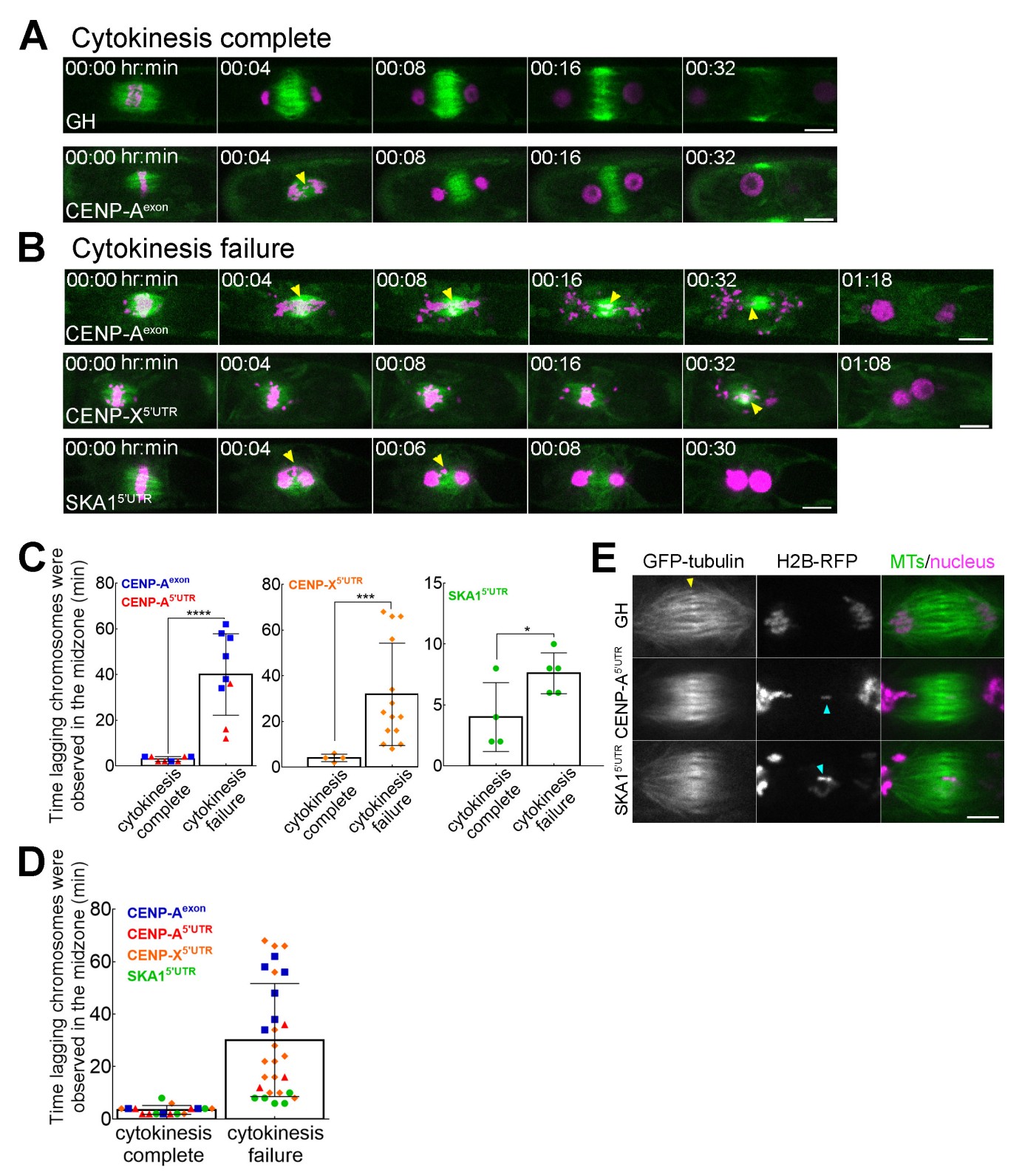

**Figure 2.** Lagging chromosomes in anaphase induce cytokinesis failure. (**A, B**) Lagging chromosomes (yellow arrowheads) present for several minutes in the midzone between separated chromatids cause cytokinesis failure in CENP-A, CENP-X and SKA1 RNAi lines. GH represents a control line. Bars, 10 µm. See *Figure 2—figure supplements 1*, *2*, *Videos 5–8*. (**C, D**) Correlation between cytokinesis failure and duration of lagging chromosomes observed in the midzone in the individual RNAi lines (**C**) and as combined data (**D**). Asterisks indicate significant differences between two groups

*Figure 2 continued on next page*

*Figure 2 continued*

(lagging chromosomes observed for short time or for several minutes) for two outcomes: cytokinesis complete and cytokinesis failure, calculated individually for CENP-A; CENP-X and SKA1 RNAi lines (*p=0.0476, ***p=0.0003, ****p<0.0001; Fisher's test; see *Figure 2—source data 1*). Each data point corresponds to a single cell. Mean ±SD are presented. (E) Representative images of the microtubule overlap in the phragmoplast in the control line (GH) and in RNAi lines (CENP-A and SKA1) with lagging chromosomes. Note that microtubule overlaps appear more broad and fuzzy in RNAi cells. Yellow arrow indicates microtubule overlaps, whereas cyan arrows point to lagging chromosomes. Images were acquired with z-stacks and a single focal plane that best shows microtubule overlaps is presented. Bar, 5 μm.

DOI: https://doi.org/10.7554/eLife.43652.015

The following source data and figure supplements are available for figure 2:

**Source data 1.** Dataset used for Fisher's test in *Figure 2C*.
DOI: https://doi.org/10.7554/eLife.43652.018
**Figure supplement 1.** Chromosome segregation defects following depletion of CENP-A, CENP-X or SKA1.
DOI: https://doi.org/10.7554/eLife.43652.016
**Figure supplement 2.** Rescue of RNAi phenotypes by ectopic expression of SKA1-Cerulean or CENP-X-Cerulean.
DOI: https://doi.org/10.7554/eLife.43652.017

essential factor for chromosome segregation in *P. patens*. In animals, it has been proposed that CENP-S and CENP-X form a complex and play an important role in outer kinetochore assembly (*Amano et al., 2009*). It is an interesting target for further investigation if plant CENP-S/CENP-X preserves such a function.

## Chromosome missegregation causes polyploidization

We observed lagging chromosomes as well as cytokinesis failure after knocking down kinetochore components. Failure in chromosome separation/segregation and cytokinesis can be caused by a single gene mutation, if the gene has multiple functions; for example, separase Rsw4 (*radially swollen4*) in *A. thaliana* is involved in sister chromatid separation, cyclin B turnover and vesicle trafficking that is required for phragmoplast formation (*Chang et al., 2003*; *Yang et al., 2011*; *Moschou et al., 2013*; *Wu et al., 2010*). By contrast, in our study, both phenotypes were observed after RNAi treatment of CENP-A, a constitutive centromeric histone protein that is unlikely to play a direct role in cytokinesis. Furthermore, the cytokinesis phenotype frequently appeared in RNAi lines targeting other six kinetochore proteins, and only when lagging chromosomes were present. Based on these data, we propose that persistent lagging chromosomes cause cytokinesis failure. Lagging chromosomes might act as physical obstacles to perturb phragmoplast microtubule amplification and/or cell plate formation. Alternatively, persistent lagging chromosomes might produce an unknown signal or induce a certain cell state that inhibits phragmoplast expansion and/or cell plate formation in order to prevent chromosome damage, reminiscent of the NoCut pathway in animal cytokinesis (*Norden et al., 2006*; *Amaral et al., 2016*). We favor the latter model, as abnormal microtubule interdigitates were

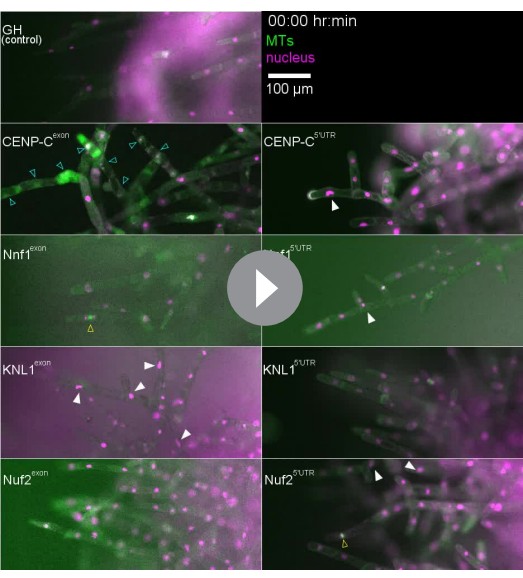

**Video 5.** Mitotic defects observed in RNAi lines targeting CENP-C, Nnf1, Nuf2 and KNL1. Representative images of mitotic progression and defects caused by depletion of four kinetochore proteins. White boxes indicate normal cell division in the control line (GH). White arrowheads show position of multinucleated cells, yellow arrowheads indicate chromosome missegregation and cytokinesis failure events, whereas cyan arrowheads show dead cells. RNAi was induced by addition of β-estradiol to the growth medium at the final concentration of 5 μM, 5–6 days prior to observation. Images were acquired at a single focal plane every 3 min. Bar, 100 μm.
DOI: https://doi.org/10.7554/eLife.43652.019

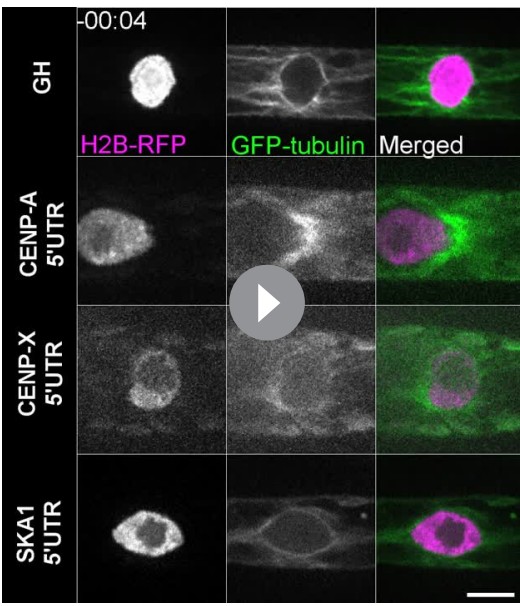

**Video 6.** Chromosome missegregation after RNAi. Representative images of mitotic progression and chromosome missegregation caused by depletion of CENP-A or CENP-X or SKA1. RNAi was induced by addition of β-estradiol to the growth medium at final concentration of 5 μM, 5–6 days prior to observation. Images were acquired at a single focal plane every 2 min. Bar, 10 μm.

DOI: https://doi.org/10.7554/eLife.43652.020

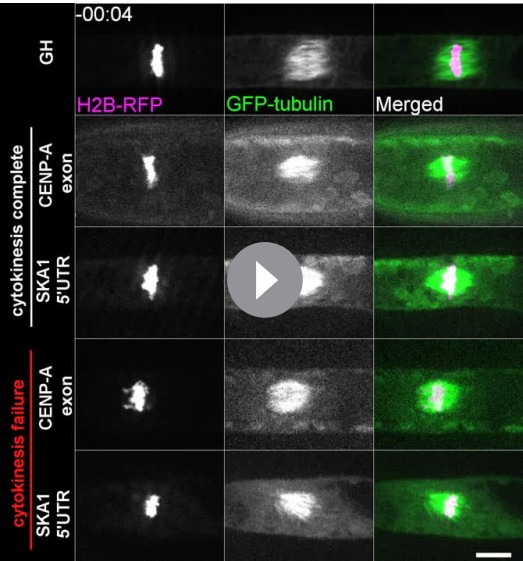

**Video 7.** Cytokinesis defect associated with lagging chromosomes in anaphase. Representative images of correlation between lagging chromosomes and cytokinesis defect in CENP-A exon RNAi and SKA1 5'UTR RNAi lines. Note that minor lagging chromosomes observed in the midzone for ≤4 min did not affect cytokinesis (*upper rows*); however, lagging chromosomes persistent for ≥6 min resulted in cytokinesis failure (*bottom rows*). This correlation is conserved in both CENP-A exon RNAi and SKA1 5'UTR RNAi lines. Cytokinesis failure was concluded when the nucleus moved without restraint of the cell plate. RNAi was induced by addition of β-estradiol to the growth medium at final concentration of 5 μM, 5–6 days prior to observation. Images were acquired at a single focal plane every 2 min. Bar, 10 μm.

DOI: https://doi.org/10.7554/eLife.43652.021

observed in the whole phragmoplast and not limited to the region proximal to the lagging chromosome (*Figure 2E*). Notably, in a recent study, cytokinesis in moss protonema cells could be completed despite longer microtubule overlaps (*de Keijzer et al., 2017*). It suggests that abnormal microtubule interdigitates represent the consequence of microtubule dynamics mis-regulation rather than the direct cause of cytokinesis failure.

Our data further suggest that, in *P. patens*, chromosome missegregation in a single cell could lead to the generation of polyploid plants. Could lagging chromosomes cause polyploidization through somatic cell lineage in wild-type plants? In our imaging of control moss cells, we could not find any lagging chromosome, since mitotic fidelity is very high in our culture conditions. Intriguingly, however, various mitotic abnormalities, including lagging chromosomes have been long observed in wild-type plants and crops, albeit at a low frequency and/or under harsh natural conditions (*Menén-dez-Yuffá et al., 2000*; *Nichols, 1941*; *Kvitko et al., 2011*). Those studies did not analyze the relationship between lagging chromosomes and cytokinesis integrity; we expect the presence of lagging chromosomes for a certain duration to similarly perturb cytokinesis as observed in our study of moss, since the cytokinesis process is highly conserved between bryophytes and angiosperms (*Smertenko et al., 2017*). Genome sequencing suggests that *P. patens*, like many other plant species, experienced whole genome duplication at least once during evolution (*Rensing et al., 2008*). Polyploidization through spontaneous mitotic errors in somatic cells might have a greater impact on de novo formation of polyploid plants than previously anticipated.

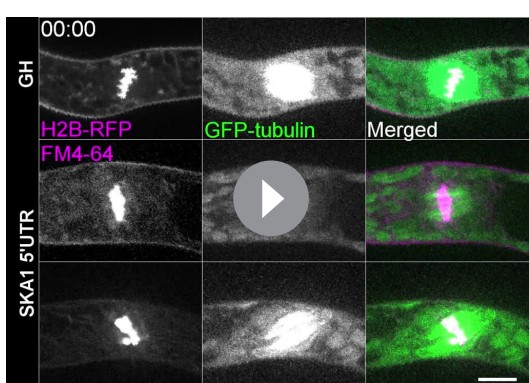

**Video 8.** Visualization of the cell plate formation using FM4-64 dye. Representative images of cytokinesis in the control GH line (*upper row*), SKA1 5'UTR RNAi line with minor lagging chromosomes (*middle row*), and with persistent lagging chromosomes (*bottom row*). Cell plate formation was visualized with 25 μM endocytic FM4-64 dye added during metaphase. FM4-64 dye was prone to photobleaching, and therefore was sometimes supplied multiple times during long-term imaging (*bottom row*). Images were acquired at a single focal plane every 2 min. Bar, 10 μm.
DOI: https://doi.org/10.7554/eLife.43652.022

## Materials and methods

### Moss culture and transformation

We generally followed protocols described by *Yamada et al. (2016)*. In brief, *Physcomitrella patens* culture was maintained on BCDAT medium at 25°C under continuous light. Transformation was performed with the polyethylene glycol-mediated method and successful endogenous tagging of the selected genes was confirmed by PCR (*Yamada et al., 2016*). We used *P. patens* expressing mCherry-α-tubulin under the pEF1α promoter as a host line, except for Mis12-mCherry line where GFP-α-tubulin line was used as a host line. For knockout, CRISPR (*Lopez-Obando, 2016*) and RNAi transformations, we used the GH line, expressing GFP-tubulin and HistoneH2B-mRFP. *P. patens* lines developed for this study are described in *Supplementary file 1*.

### Plasmid construction

Plasmids and primers used in this study are listed in *Supplementary file 2*. For the C-terminal tagging, we constructed integration plasmids, in which ~800 bp C-terminus and ~800 bp 3'-UTR sequences of the kinetochore gene were flanking the *citrine* gene, the nopaline synthase polyadenylation signal (nos-ter), and the G418 resistance cassette. For the N-terminal tagging we constructed integration plasmids, in which ~800 bp 5'-UTR and ~800 bp N-terminus sequences of the kinetochore gene were flanking the *citrine* gene. CENP-A cDNA was amplified by PCR and subcloned into a vector containing the rice actin promoter, *citrine* gene, the rbcS terminator, the modified *aph4* cassette, and flanked by the genomic fragment of the *hb7* locus to facilitate integration. All plasmids were assembled with the In-Fusion enzyme according to manufacturer's protocol (Clontech). RNAi constructs were made by using the Gateway system (Invitrogen) with pGG624 as the destination vector (*Miki et al., 2016*).

### DNA staining

We followed the protocol described by *Vidali et al. (2007)* with the following modifications: sonicated moss was cultured for 6–7 days on the BCDAT plate, containing 5 μM β-estradiol for RNAi induction and 20 μg/ml G418 to prevent contamination. Collected cells were preserved in a fixative solution (2% formaldehyde, 25 mM PIPES, pH 6.8, 5 mM MgCl$_2$, 1 mM CaCl$_2$) for 30 min and washed three times with PME buffer (25 mM PIPES, pH 6.8, 5 mM MgCl$_2$, 5 mM EGTA). Following fixation, cells were mounted on 0.1%PEI (polyethyleneimine)-coated glass slides and subsequently incubated with 0.1% Triton X-100 in PME for 30 min and 0.2% driselase (Sigma-Aldrich) in PME for 30 min. Next, cells were washed twice in PME, twice in TBS-T buffer (125 mM NaCl, 25 mM Tris-HCl, pH 8, and 0.05% Tween 20) and mounted in 10 μg/mL DAPI in TBS-T for observation. Images were acquired with the Olympus BX-51 fluorescence microscope equipped with ZEISS Axiocam 506 Color and controlled by ZEN software. Fluorescent intensity was measured with ImageJ. Cytoplasmic background was subtracted.

### Live-imaging microscopy

A glass-bottom dish (Mattek) inoculated with moss was prepared as described in *Yamada et al. (2016)* and incubated at 25°C under continuous light for 4–7 days before live-imaging. To observe RNAi lines, we added 5 μM β-estradiol to culture medium (*Miki et al., 2016*). For the high-magnification time-lapse microscopy, the Nikon Ti microscope (60 × 1.40 NA lens or 100 × 1.45 NA lens)

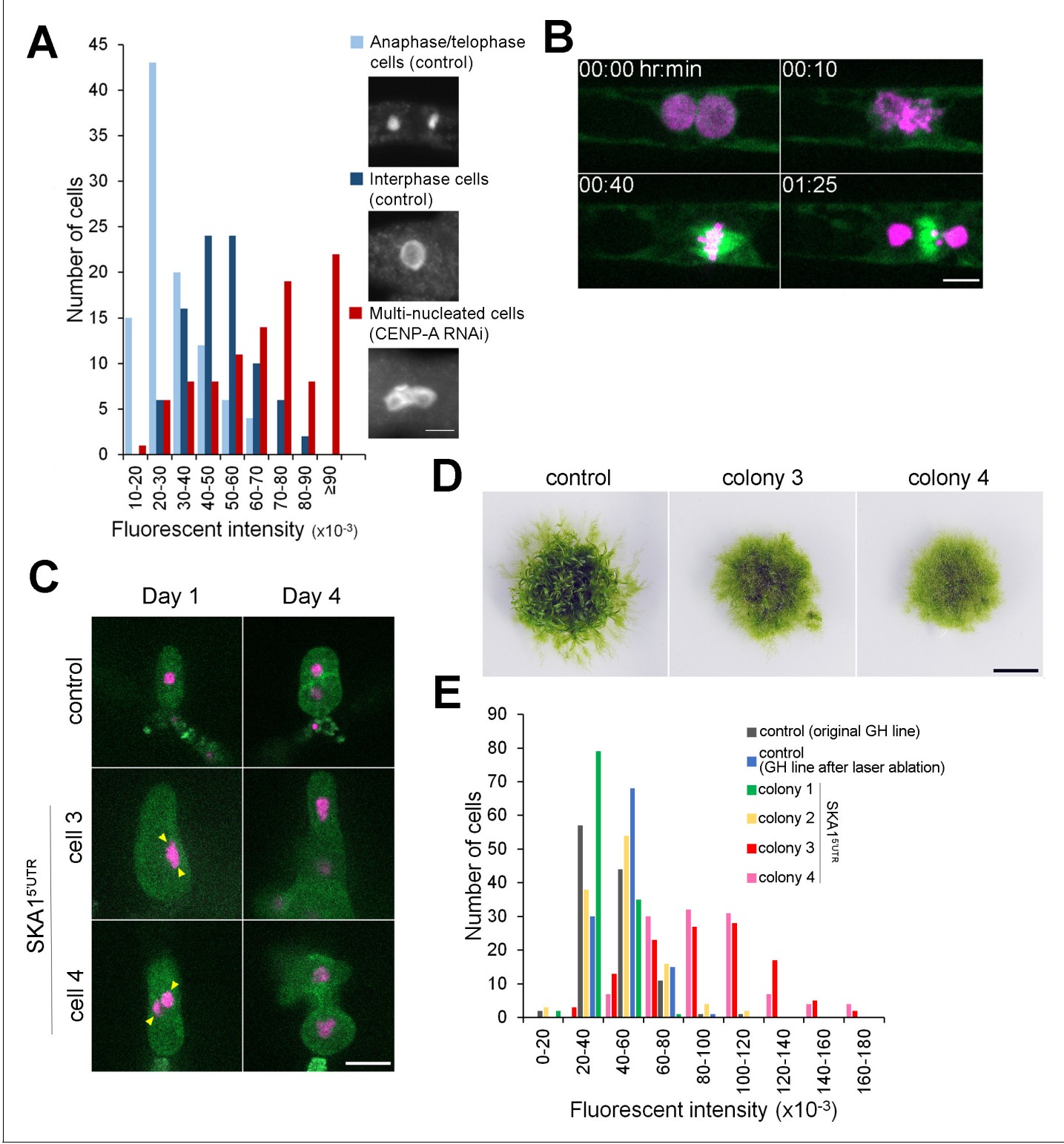

**Figure 3.** Cytokinesis failure in somatic cells can generate plants with whole-genome duplication. (**A**) Quantification of the nuclear DNA content. Anaphase/telophase cells were used as a standard for 1N nuclei (light blue). Interphase cells randomly selected in the control line mostly had double amounts of DNAs as expected (dark blue), whereas cells that failed cytokinesis had higher ploidy (red). DNA amounts are shown as fluorescent intensity of the DAPI-stained nuclei per cell after subtraction of the cytoplasmic background. (**B**) Representative images of mitotic entry and single spindle formation of the multi-nucleated cell in the *P. patens* SKA1 RNAi line. Bar, 5 µm. See *Video 9*. (**C**) Regeneration of a single cell isolated by laser dissection microscopy from the control cell line (GH) or multi-nucleated cells from SKA1 RNAi line (multi-nuclei are marked with yellow arrowheads).
*Figure 3 continued on next page*

*Figure 3 continued*

Bar, 50 μm. (**D**) Moss colonies regenerated from single cells. Bar, 0.5 cm. (**E**) Quantification of the nuclear DNA content in the interphase nucleus of regenerated moss colonies, corresponding to (**C**) and (**D**).

DOI: https://doi.org/10.7554/eLife.43652.023

equipped with the spinning-disk confocal unit CSU-X1 (Yokogawa) and an electron-multiplying charge-coupled device camera (ImagEM; Hamamatsu) was used. Images were acquired every 30 s for localization analysis and every 2 min for RNAi analysis. The microscope was controlled by the Micro-Manager software and the data was analyzed with ImageJ. The rescue lines for RNAi were observed using a fluorescence microscope (IX-83; Olympus) equipped with a Nipkow disk confocal unit (CSU-W1; Yokogawa Electric) controlled by Metamorph software.

## Single-cell isolation

Protonema tissue of *P. patens* was sonicated, diluted with BCD medium with 0.8% agar, and spread on cellophane-covered BCDAT plates that contain 5 μM estradiol to induce RNAi. After 5–6 days, small pieces of cellophane containing clusters of protonemal cells (each containing 3–20 cells) were cut with scissors and placed upside-down on a glass-bottom dish. Bi- or multi-nucleated cells were identified using Axio Zoom.v16. Single bi-nucleated cell (SKA1 RNAi line) or random cell (control GH line) was selected for isolation, and all other cells were ablated with a solid-state ultraviolet laser (355 nm) through a 20X objective lens (LD Plan-NEOFLUAR, NA 0.40; Zeiss) at a laser focus diameter of less than 1 μm using the laser pressure catapulting function of the PALM microdissection system (Zeiss). Irradiation was targeted to a position distantly located from the cell selected for isolation to minimize the irradiation effect. Note that visual distinction of multi-nucleated cells from those with slightly deformed nuclei is not easy in *P. patens*, since in multi-nucleated cells, the nuclei maintain very close association with each other, so that nuclear boundaries often overlap. We interpret that two of four regenerated protonemata had haploid DNA content due to our unintentional isolation of a single cell with a deformed nucleus rather than multi-nuclei. Next, a piece of cellophane with single isolated cell was transferred from the glass-bottom dish to estradiol-free medium (20 μg/ml G418 was supplied to prevent bacterial/fungal contamination). DAPI staining was performed 5–6 weeks later as described above.

## Sequence analysis

Full-size amino acid sequences of the selected proteins were aligned using MAFFT ver. 7.043 and then revised manually with MacClade ver. 4.08 OSX. We used the Jones-Taylor-Thornton (JTT) model to construct maximum-likelihood trees in MEGA5 software. Statistical support for internal branches by bootstrap analyses was calculated using 1000 replications. Reference numbers correspond to Phytozome (www.phytozome.net) for *Physcomitrella patens*, the Arabidopsis Information Resource (www.arabidopsis.org) for *Arabidopsis thaliana* and Uniprot (www.uniprot.org) for *Homo sapiens*. Original protein alignments after MAFFT formatted with BoxShade (https://embnet.vital-it.ch/software/BOX_form.html) are shown in *Supplementary file 3*.

## Acknowledgements

We are grateful to Dr. Yoshikatsu Sato and Nagisa Sugimoto for their assistance with laser ablation experiments; to Dr. Peishan Yi, Moé Yamada and Shu Yao Leong for comments and discussion; and Rie Inaba for technical assistance. Imaging was partly conducted in the Institute of Transformative Bio-Molecules (WPI-ITbM)

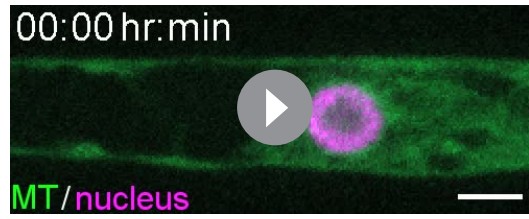

**Video 9.** Mitotic entry of the multi-nucleated cell in *P. patens*. SKA1 5'UTR RNAi was induced by addition of β-estradiol to the growth medium at final concentration of 5 μM, 5–6 days prior to observation. Multi-nucleated cells resulting from cytokinesis failure were monitored with the spinning-disk confocal microscope. Images were acquired at a single focal plane every 5 min. Bar, 10 μm.

DOI: https://doi.org/10.7554/eLife.43652.024

at Nagoya University, supported by Japan Advanced Plant Science Network. This work was funded by JSPS KAKENHI (17H06471, 17H01431) to GG The authors declare no competing financial interests.

## Additional information

### Funding

| Funder | Grant reference number | Author |
|---|---|---|
| Japan Society for the Promotion of Science | KAKENHI 17H06471 | Gohta Goshima |
| Japan Society for the Promotion of Science | KAKENHI 17H01431 | Gohta Goshima |

The funders had no role in study design, data collection and interpretation, or the decision to submit the work for publication.

### Author contributions

Elena Kozgunova, Conceptualization, Resources, Data curation, Formal analysis, Investigation, Writing—original draft; Momoko Nishina, Resources, Investigation, Methodology; Gohta Goshima, Conceptualization, Funding acquisition, Writing—original draft

### Author ORCIDs

Elena Kozgunova (iD) http://orcid.org/0000-0002-6854-2071
Gohta Goshima (iD) http://orcid.org/0000-0001-7524-8770

### Decision letter and Author response

Decision letter https://doi.org/10.7554/eLife.43652.030
Author response https://doi.org/10.7554/eLife.43652.031

## Additional files

### Supplementary files

• Supplementary file 1. *Physcomitrella patens* transgenic lines generated in this study.
DOI: https://doi.org/10.7554/eLife.43652.025

• Supplementary file 2. Plasmids and primers used in this study.
DOI: https://doi.org/10.7554/eLife.43652.026

• Supplementary file 3. Protein alignments used for the phylogeny analysis.
DOI: https://doi.org/10.7554/eLife.43652.027

• Transparent reporting form
DOI: https://doi.org/10.7554/eLife.43652.028

### Data availability

All data generated or analysed during this study are included in the manuscript and supporting files.

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
