## [Decision Letter]

Thank you for submitting your article "Chromosome missegregation causes somatic polyploidization in plants" for consideration by *eLife*. Your article has been reviewed by two peer reviewers, and the evaluation has been overseen by a Reviewing Editor and Christian Hardtke as the Senior Editor. The reviewers have opted to remain anonymous.

The reviewers have discussed the reviews with one another and the Reviewing Editor has drafted this decision to help you prepare a revised submission.

Summary:

This manuscript addresses the functions of kinetochore proteins in the moss *P. patens*. 18 putative kinetochore proteins in this moss were identified on the basis of their homology with known proteins from human, then tagged versions were localized to examine how they behaved over the cell cycle. Several candidate proteins localized to the kinetochores in *P. patens*. There are a few minor surprises: CENPC is not constitutive, the rest of the CCAN does not localize, and KNLI and SKA1 do localize to kinetochores. They then examine phenotypes of RNAi knockdowns for 3 proteins and find lagging chromosomes, and show that polyploid cells (i.e., diploid from haploid) cells arose from such lines. The authors propose that lagging chromosomes inhibit phragmoplast functions and cell plate assembly, then propose that similar delays of chromosome segregation could have been a mechanism for natural polyploidization during plant evolution.

Title: The title is overly broad and emphasizes the speculation about polyploidization (using the strong word "causes") rather than the data in the paper, which is about one organism (not plants in general) and about the localization of some (not all) kinetochore proteins and the consequences of RNAi knockdowns for a few of these. Please revise the title to include *Physcomitrella* in the title, and emphasize the results in the paper.

Essential revisions:

1) It is not clear from the text why localization of Nnf1, Dsn1, Spc24, SKA3 was not examined. Why were CENPX and SKA1 chosen? Why avoid the obvious choices, i.e. CENPC, MIS12, and NDC80? The authors should provide some language and preferably data to show what happened with those traditional kinetochore proteins. It is extremely tenuous to use CENPX to support the CENPA results given that their CENPX-tagged version did not localize to kinetochores. It is very strange that they chose SKA1 as an outer kinetochore protein and not NDC80. It seems like something is being avoided here. Please list all the genes targeted by RNAi and whether all of them showed laggards and polyploidy, not just the select few that support their polyploidy claim.

2) Please address whether the chromosome lagging and the cytokinetic failure could be partly consequence of failed microtubule stabilization in the central spindle, rather than solely a phragmoplast problem. The phenotypes resemble phenotypes seen after treatment with microtubule drugs. It appears that the phragmoplast (with classical anti-parallel microtubule overlap in the midzone) does not form. The quality of the images and videos is not sufficient to determine the reason for the cytokinetic failure. For example, Video 6 shows lack of cell plate initiation in the SKA1 knockdown. It is possible that cell division fails due to the abnormal anaphase B spindle morphology? If kinetochore failures cause major changes in microtubule organization, perhaps this better explains why CENPX, wherever it functions, had the same basic phenotype as CENH3?

3) According to the localization data, kinetochore protein composition varies as cells progress through the cell cycle. It is not clear why the structure of the kinetochore or the stoichiometry of kinetochore proteins changes. Could it be an artifact of tagging? Only one position of the tag was tested. The localization pattern could change if the tag was used at the other end of the protein. Please address these points in the revision.

4) Regarding the RNAi knockdowns. Why this unlikely trio, CENPA, CENPX, and SKA1? In each of these cases, they find that severe lagging (greater than 12 min, but not less than 4 min) leads to restitution and reformation of diploid nuclei. The numbers are not large here and the phenotypes for each mutant are a little different. However, they make the point that in all cases, no matter what the origin, long-delayed lagging chromosomes led to cells with 2 nuclei and genome doubling (from haploid to diploid).

5) The idea that kinetochore defects can lead to polyploidy is appealing and that idea is largely presented as theory and not necessarily as fact (except in the title). However, the extrapolation of their results to angiosperms should be tempered. The 1938 McClintock paper is on ring chromosomes which form bridges following sister chromatid exchange (they are bridges, not laggards). It is inaccurate to cite this as evidence that as much as 8% of plant mitoses have errors, since these were special mutant lines. Likewise, they cite studies involving tissue culture (which is known to induce segregation errors), and old seeds, and plants from extreme environments. Please reword to say errors have been found in mutant lines and extreme environments (with citations) and might also occur in normal unperturbed tissues at some frequency. You don't need 8% to make the story. Polyploidy events occur over evolutionary timescales.

---

## [Author Response]

Title: The title is overly broad and emphasizes the speculation about polyploidization (using the strong word "causes") rather than the data in the paper, which is about one organism (not plants in general) and about the localization of some (not all) kinetochore proteins and the consequences of RNAi knockdowns for a few of these. Please revise the title to include Physcomitrella in the title, and emphasize the results in the paper.

We have modified the title following this advice and based it on the actual data, including those newly acquired during revision.

Essential revisions:1) It is not clear from the text why localization of Nnf1, Dsn1, Spc24, SKA3 was not examined.

It is established that kinetochores are composed of several protein sub-complexes, including the Mis12 complex, Ndc80 complex, and SKA complex (Musacchio and Desai, 2017). Initially, we picked up at least one subunit per sub-complex as representatives (Figure 1A and Figure 1—figure supplements 1-7). It is likely that other components of a sub-complex behave similarly, as they are strongly bound to each other. Consistent with this notion, we have shown the identical localisation pattern of Mis12 and Nnf1; Ndc80, Nuf2 and Spc25; SKA1 and SKA2 as members of the Mis12, Ndc80 and SKA complex, respectively (the Nnf1-Citrine data on Figure 1B and Figure 1—figure supplement 5B is a new addition to the revised manuscript). We have also provided a more detailed explanation in the text regarding our selection strategy for gene tagging.

Why were CENPX and SKA1 chosen? Why avoid the obvious choices, i.e. CENPC, MIS12, and NDC80? The authors should provide some language and preferably data to show what happened with those traditional kinetochore proteins. It is extremely tenuous to use CENPX to support the CENPA results given that their CENPX-tagged version did not localize to kinetochores. It is very strange that they chose SKA1 as an outer kinetochore protein and not NDC80. It seems like something is being avoided here. Please list all the genes targeted by RNAi and whether all of them showed laggards and polyploidy, not just the select few that support their polyploidy claim.

We tried to establish multiple kinetochore protein RNAi lines and obtained CENP-A, SKA1 and CENP-X lines first; therefore, we focused our analysis on those lines. However, since manuscript submission, we have also selected RNAi lines and performed time-lapse microscopy for CENP-C, Nnf1, KNL1 Ndc80, Nuf2, and borealin. The complete list, along with brief phenotype descriptions and accompanying notes, has been presented in Figure 1—figure supplement 1. Representative image sequences of several new RNAi lines can be found in Video 5.

RNAi lines targeting 7 out of 10 genes frequently showed phenotypes similar to CENP-A/SKA1/CENP-X, namely chromosome missegregation, multi-nucleated cells, and/or cytokinesis failure (new data of CENP-C, Nnf1, KNL1 and Nuf2 are shown in Video 5). In the remaining Ndc80 and borealin RNAi lines, we have also observed multi-nucleated cells and cytokinesis failure, but the frequency of cytokinesis failure was low compared to other RNAi, possibly due to incomplete RNAi; therefore, we did not include this data in the manuscript. Finally, we could not obtain a Mis12 RNAi line with mitotic phenotypes even after multiple attempts (Nakaoka et al., 2012; two more unpublished attempts); RNAi lines cannot always be selected in our hands. In the case of Mis12, a small amount of protein might be sufficient to fulfil its functions, consequently giving rise to perceived RNAi resistance (this was the case for yeast Mis12; Goshima et al. EMBO J. 2003). More importantly, we did see the expected phenotypes of chromosome missegregation and cytokinesis failure for Nnf1, another member of the Mis12 complex.

Overall, the high probability of phenotype detection has reinforced our conclusion that chromosome missegregation underlies cytokinesis failure and polyploidization in *P. patens*.

2) Please address whether the chromosome lagging and the cytokinetic failure could be partly consequence of failed microtubule stabilization in the central spindle, rather than solely a phragmoplast problem. The phenotypes resemble phenotypes seen after treatment with microtubule drugs. It appears that the phragmoplast (with classical anti-parallel microtubule overlap in the midzone) does not form. The quality of the images and videos is not sufficient to determine the reason for the cytokinetic failure. For example, Video 6 shows lack of cell plate initiation in the SKA1 knockdown. It is possible that cell division fails due to the abnormal anaphase B spindle morphology? If kinetochore failures cause major changes in microtubule organization, perhaps this better explains why CENPX, wherever it functions, had the same basic phenotype as CENH3?

To address this point, we have conducted a high resolution Z-stack imaging in the control, CENP-A, and SKA1 RNAi lines. In 5 out of 7 control cells, we observed sharp microtubule overlaps in the phragmoplast (Figure 2E; Hiwatashi et al., 2008). Microtubules signals were also enriched in phragmoplast midzone of the RNAi cells; however, the overlap was fuzzy and less prominent compared to the control (n = 6 out of 6 SKA1 RNAi; 6 out of 6 CENP-A RNAi; 2 out of 7 control cells). These results suggest that lagging chromosomes affect microtubule organisation during telophase.

In this experiment, we noticed that microtubule overlaps are overall abnormal, irrespective of the proximity to the lagging chromosome, in the RNAi lines. This suggests that lagging chromosomes do not simply act as physical obstacles during cell plate deposition; instead, they produce a certain signal to prevent proper phragmoplast formation. This point has been discussed in the revised manuscript.

We thank the reviewers for this observation and bringing it to our attention.

3) According to the localization data, kinetochore protein composition varies as cells progress through the cell cycle. It is not clear why the structure of the kinetochore or the stoichiometry of kinetochore proteins changes. Could it be an artifact of tagging? Only one position of the tag was tested. The localization pattern could change if the tag was used at the other end of the protein. Please address these points in the revision.

Endogenous N-terminal tagging is technically more challenging than C-terminal tagging in moss (Yamada et al., 2016). We therefore tagged Citrine or mCherry to the C-terminus of 18 genes that are very likely essential for moss’s viability (CENP-C, -X, Nnf1, Ndc80, Nuf2, borealin, Mis12, KNL1 and SKA1; essentiality is assumed by failure of KO selection as well as robust RNAi phenotype). Since the “replacement” lines grew indistinguishably from the wild-type, we naturally interpreted that the tagging did not perturb the protein function (note that protonemal tissue is haploid; hence, when no orthologues are present, no other endogenous gene is left in the replacement line).

We think that the localisation of the function-verified and tagged protein represents the endogenous localisation well; in fact, this is a standard approach in yeast, where homologous recombination has been routinely applied, and, more recently, in animal tissue culture cells, for which endogenous tagging becomes possible owing to CRISPR technology.

Nevertheless, we also selected both N- and C-terminal Citrine lines for CENP-C and CENP-O, and verified the identical localisation patterns (newly added Video 3). C-terminal tagging line of CENP-S and N-terminal tagging line of CENP-X could not be selected after two attempts, suggesting that, in those two cases, tagging perturbed protein function.

For CENP-O, KNL2 (Mis18BP1), Ndc80, Spc25, SKA2, BubR1 and Mps1 genes, the functionality of the Citrine tagging could not be confirmed, since there is an additional orthologue present in the genome.

We have summarised our attempts and data on Citrine/mCherry tagging in the Figure 1—figure supplement 1 of the revised manuscript.

4) Regarding the RNAi knockdowns. Why this unlikely trio, CENPA, CENPX, and SKA1? In each of these cases, they find that severe lagging (greater than 12 min, but not less than 4 min) leads to restitution and reformation of diploid nuclei. The numbers are not large here and the phenotypes for each mutant are a little different. However, they make the point that in all cases, no matter what the origin, long-delayed lagging chromosomes led to cells with 2 nuclei and genome doubling (from haploid to diploid).

We believe this is the same comment as comment 1. Please refer to our responses above.

5) The idea that kinetochore defects can lead to polyploidy is appealing and that idea is largely presented as theory and not necessarily as fact (except in the title). However, the extrapolation of their results to angiosperms should be tempered. The 1938 McClintock paper is on ring chromosomes which form bridges following sister chromatid exchange (they are bridges, not laggards). It is inaccurate to cite this as evidence that as much as 8% of plant mitoses have errors, since these were special mutant lines. Likewise, they cite studies involving tissue culture (which is known to induce segregation errors), and old seeds, and plants from extreme environments. Please reword to say errors have been found in mutant lines and extreme environments (with citations) and might also occur in normal unperturbed tissues at some frequency. You don't need 8% to make the story. Polyploidy events occur over evolutionary timescales.

We thank the reviewers for pointing out this. We have modified the text accordingly.